# Decision Rules are in the Pixels: Towards Pixel-level Evaluation of Saliency-based XAI Models

## Abstract

The intricate and opaque nature of deep neural networks (DNNs) makes it difficult to decipher how they make decisions. Explainable artificial intelligence (XAI) has emerged as a promising remedy to this conundrum. However, verifying the correctness of XAI methods remains challenging, due to the absence of universally accepted ground-truth explanations. In this study, we focus on assessing the correctness of saliency-based XAI models applied to DNN-based image classifiers *at the pixel level*. The proposed evaluation protocol departs significantly from previous human-centric correctness assessment at the semantically meaningful object part level, which may not correspond to the actual decision rules derived by classifiers. A crucial step in our approach involves introducing a spatially localized shortcut to the image, a form of decision rule that DNN-based classifiers tend to adopt preferentially, without disrupting original image patterns and decision rules therein. After verifying the shortcut as the dominant decision rule, we estimate the Shapley value for each pixel within the shortcut area to generate the ground-truth explanation map, assuming that pixels outside this area have null contributions. We quantitatively evaluate fourteen saliency-based XAI methods for classifiers utilizing convolutional neural networks and vision Transformers, trained on perturbed CIFAR-10, CIFAR-100, and ImageNet datasets, respectively. Comprehensive experimental results show that existing saliency-based XAI models struggle to offer accurate pixel-level attributions, casting doubt on the recent progress in saliency-based XAI.

## 1 Introduction

Machine learning models, particularly in the deep learning era, have demonstrated remarkable performance across a wide range of engineering applications (Redmon et al., 2016; Long et al., 2015; Vinyals et al., 2015; Antol et al., 2015). Yet, their intrinsic complexity and opacity present significant challenges, especially when it comes to interpreting their predictive outcomes (Erhan et al., 2009). Insights into the decision-making processes of DNNs are crucial in fields like healthcare, finance, and law, where understanding the rationale behind a decision can be as important as the decision itself, and for industries governed by regulations that demand transparency (Samek et al., 2017; Molnar, 2020; Borys et al., 2023). Additionally, interpretable models can foster trust, ensure fairness, improve human-artificial intelligence (AI) collaboration, and be more easily rectified. Due to these reasons, numerous efforts have been made to demystify the behaviors of machine learning models, especially deep neural networks (DNNs), through the development of *post-hoc* explainable AI (XAI) models (Linardatos et al., 2020; Ali et al., 2023).

While the rapid advancement of XAI models is commendable, the conflicting explanations they often produce highlight the need for a reliable evaluation protocol to assess their correctness before we can derive additional information from such explanations (Nauta, 2023). Evaluating XAI models, however, remains a significant challenge due to the lack of ground-truth explanations. In the context of saliency-based XAI models designed for image classifiers (Zeiler & Fergus, 2014; Ribeiro et al., 2016; Sundararajan et al., 2017; Petsiuk et al., 2018), the predominant evaluation approach involves feature deletion, under the assumption that each feature operates independently. This approach disregards the complex interdependencies among features, which can result in inaccurate estimation

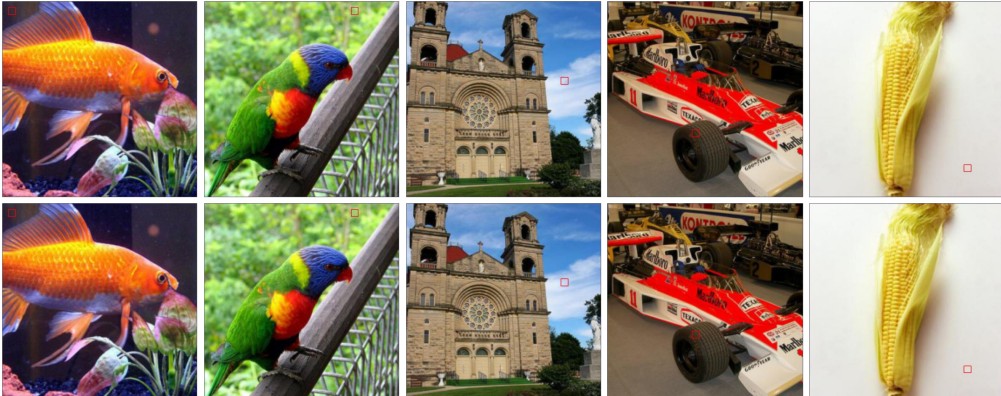

Figure 1: Minimal perturbed ImageNet. Top: clean image examples for 5 categories; Bottom: corresponding perturbed images with perturbed patches as shortcuts. The area highlighted by the red box indicates the difference between the image pairs.

of feature importance and, consequently, flawed evaluations (Chen et al., 2019; Hesse et al., 2023; Alvarez Melis & Jaakkola, 2018; Hesse et al., 2024). Additionally, when a feature is removed, a placeholder (*e.g.*, an all-zero mask) is commonly inserted to maintain the input dimension required by image classifiers. However, this substitution may introduce class-conditioned information (*e.g.*, the contour of the mask), and thus create unintended decision rules to cause unexpected classifier behaviors (Selvaraju et al., 2017b; Samek et al., 2016; Hooker et al., 2019; Rong et al., 2022). Moreover, to keep computational complexity manageable and to align with human-centric explanations, features are typically defined at the level of object parts (Hesse et al., 2023) or image patches (Hesse et al., 2024), which sacrifices the opportunity for more granular pixel-level assessment.

We argue that assessing saliency-based XAI models at the pixel level is vital as there is evidence that altering even a single pixel can drastically change a classifier's predictions (Su et al., 2019). In pursuit of this, we introduce a new evaluation protocol for saliency-based XAI models by training classifiers on natural image datasets that have been locally perturbed (Sadasivan et al., 2023). Our evaluation protocol relies on a shortcut decision rule that offers three beneficial properties. First, classifiers often prioritize this added shortcut as the dominant decision rule, overshadowing existing known or unknown rules in the image. This can be substantiated through a straightforward computational sanity check. In contrast, previous synthetic datasets (Oramas et al., 2017; Hesse et al., 2023) used to evaluate XAI methods typically contain a unique decision rule in the image, making them less realistic. Second, the added shortcut has a minimal impact on the decision rules (patterns) in the original unperturbed dataset[1]. This enables sample-efficient training of classifiers on the perturbed dataset alone, unlike the FunnyBirds dataset (Hesse et al., 2023), which requires a combinatorial number of mixtures of original and feature-deleted images to ensure stable classifier behavior. Third, the shortcut is highly spatially localized (see Fig. 1), facilitating the derivation of pixel-level ground-truth explanations.

To approximately derive pixel-level ground-truth explanations, we compute the Shapley value (Shapley, 1953) of each pixel in the shortcut area as a way of measuring pixel importance. Unlike the previous feature deletion methods with feature independence assumption, our method considers all possible joint effects of shortcut pixels while assuming that pixels outside the shortcut area have null contributions (*i.e.*, zero importance scores), when the shortcut is verified to be the dominant decision rule. We assess the correctness of fourteen saliency-based XAI methods applied to four DNN-based classifiers (*i.e.*, VGG-16 (Simonyan & Zisserman, 2014), ResNet-50 (He et al., 2016), ViT-B (Dosovitskiy et al., 2020) and SWinT-B (Liu et al., 2021)) trained on three widely adopted image datasets (*i.e.*, CIFAR-10 (Krizhevsky & Hinton, 2009), CIFAR-100 (Krizhevsky & Hinton, 2009), and ImageNet (Russakovsky et al., 2015)). Our findings consistently indicate that current XAI methods struggle to provide accurate pixel-level explanations, thereby casting doubt on the recent progress of saliency-based XAI.

---

[1]This is empirically demonstrated by the consistent performance of classifiers trained on the original dataset and tested on the perturbed dataset.

## 2 RELATED WORK

### 2.1 SALIENCY-BASED XAI METHODS

In the realm of XAI, numerous methods have been proposed to interpret and elucidate the predictions yielded by DNNs (Nauta, 2023). This study focuses on post-hoc saliency-based XAI methods, which can be generally classified into three categories: gradient-based approaches, perturbation-based approaches and attention-based approaches (Borys et al., 2023; Kokhlikyan et al., 2020; Agarwal et al., 2021). Gradient-based methods utilize the gradient of a classifier's output associated with its input to generate explanations, such as Gradient (Simonyan et al., 2014) and Integrated Gradients (IG) (Sundararajan et al., 2017). However, they have been observed to lack robustness and are sensitive to factors that have no contribution to the model's decision (Ghorbani et al., 2019; Kindermans et al., 2019). On the other hand, perturbation-based techniques, including RISE (Petsiuk et al., 2018) and LIME (Ribeiro et al., 2016), are effective and easy to implement, involve perturbing the input and monitoring the subsequent effect on the model's output. Nonetheless, they often encounter challenges with unexpected model behaviors caused by masking operations. Most saliency-based methods (Selvaraju et al., 2017b; Zeiler & Fergus, 2014) are initially designed for convolutional neural networks (CNNs), while for Transformers (Dosovitskiy et al., 2020), attention-based methods like Attention Rollout (AR) (Abnar & Zuidema, 2020) aggregate the attention maps within the model to generate saliency maps, thereby enhancing interpretability to attention mechanism.

Notably, GradientSHAP, a gradient-based method, and KernelSHAP, a perturbation-based method, both utilize Shapley value estimation (Lundberg & Lee, 2017) to ensure a fair allocation of feature contributions. However, the computation load can increase exponentially with the number of features, making it a significant challenge to estimate the Shapley value efficiently. Existing methods are typically surrogate and estimate the value at a superpixel level (Lundberg & Lee, 2017; Jethani et al., 2021).

### 2.2 XAI CORRECTNESS EVALUATION

Although most saliency-based XAI methods do offer pixel attributions, the attributions are inconsistent, and there is a notable absence of standardized and reliable evaluation protocols. In the existing literature, evaluating explanation correctness for model behavior is typically conducted through simulations on synthetic datasets or feature deletion techniques (Nauta et al., 2023). Oramas et al. (2017) and Hesse et al. (2023) created synthetic image datasets using stacked object blocks with discriminative attributes such as shape, color, or position. However, they typically contain a unique decision rule, which is less realistic. Other methods (Adebayo et al., 2020; Ross et al., 2017; Lin et al., 2021) train the classifier to learn introduced shortcuts, treating them as ground truth explanations to conduct evaluation. However, the ground truths they provide are often binary masks that highlight a patch or the entire image background, which are too coarse to accurately represent the true pixel attributions.

Another typical approach is to remove features from the input and observe the effect on the output logit or prediction accuracy. Features can be removed individually (single deletion) (Alvarez Melis & Jaakkola, 2018; Chen et al., 2019; Selvaraju et al., 2017b; Hesse et al., 2023; 2024) or in an iterative manner (incremental deletion) (Hooker et al., 2019; Rong et al., 2022; Samek et al., 2016). However, a single deletion protocol may not consider the combinations and orders between different features and assume features are independent, potentially decreasing evaluation reliability. In addition, deleting features, for instance, replacing them with constant (*e.g.*, zero or dataset mean), may introduce new patterns, *i.e.*, the placeholder features can cause unexpected classifier behaviors. Training the classifier on a mix of clean and perturbed images may not solve this (Hooker et al., 2019; Hesse et al., 2023), as an unintended shortcut(*e.g.*, the contour of the mask) (Rong et al., 2022) can be introduced by the perturbation to affect training process. In addition, Hooker et al. (2019) needs to retrain the classifier for each explanation method, coming at the cost of heavy computational load. To avoid the perturbation causing information leakage, Rong et al. (2022) propose a class-independent masking mechanism by pixel interpolation.

To the best of our knowledge, our proposed protocol first provides pixel-level ground truth rather than a binary mask highlighting a patch, an object, or the whole background. We compute pixel importance using Shapley value estimation, which comprehensively considers all possible joint effects

of pixels against the assumption of feature independence. Besides, the added shortcut has a minimal impact on the decision rules (patterns) in the original clean dataset, therefore, when removing the shortcut pixels (*i.e.*, recovering the clean image) for computing pixel contribution, we can avoid introducing new patterns to cause unexpected classifier behaviors.

## 3 EVALUATION PROTOCOL

We seek to formulate the general problem of evaluating saliency-based XAI models at a pixel level for DNN-based image classifiers. Specifically, we start with a clean image dataset $\mathcal{D}$, upon which we define a class label $c \in \mathcal{C}$ for every $x \in \mathcal{D}$, where $\mathcal{C}$ is the set of categories. We locally perturb the dataset to add a shortcut decision rule. An image classifier $f : \mathbb{R}^N \mapsto \mathbb{R}^{|\mathcal{C}|}$ trained on the perturbed version of $\mathcal{D}$ can be specified to output a probability vector $p$. We consider a set of XAI models $\mathcal{G} = \{g^{(i)}\}_{i=1}^M$, with each model $g^{(i)}$ generating pixel attributions for predictions made by the classifier. Our goal is to evaluate the performance of $M$ XAI models in accurately attributing the importance of image pixels. This process involves checking dominance, estimating the ground-truth pixel importance for each image satisfying the dominant prerequisites, predicting pixel attributions using various XAI models, and then computing metrics to quantitatively evaluate the correctness of these predictions.

### 3.1 LOCAL PERTURBATION

Locally perturbing the clean dataset involves applying a $\sqrt{K} \times \sqrt{K}$ category-specific convolutional kernel to blur a $\sqrt{D} \times \sqrt{D}$ patch at a particular location. The combinations of kernels and locations are uniquely associated with each image category. We define the operation as a shortcut function $h : \mathbb{R}^N \times \mathcal{C} \mapsto \mathbb{R}^N$, and following Sadasivan et al. (2023), we apply convolutional filters $k^{(c)} \in \mathbb{R}^K$ for each class $c \in \mathcal{C}$. A random filter weight, out of the $K$ weights, in each $k^{(c)}$ is set to yield a value of 1. The rest of the filter weights are randomly drawn from a uniform distribution $\mathcal{U}[0, \alpha]$, where $\alpha$ is the blur parameter. With appropriate parameters to subtly perturb the clean image, the introduced shortcut serves as a decision rule that classifiers tend to adopt and has a minimal impact on existing rules, as exemplified in Figure 1.

### 3.2 PIXEL IMPORTANCE ESTIMATION

**Dominant Prerequisites**. The perturbed dataset undergoes slight modifications and contains numerous decision rules that can aid in prediction, including the introduced shortcut itself. However, fully comprehending the complete decision rules the classifier has learned can be highly challenging due to the inherent opaque of most machine learning models. In this context, we present two prerequisites that can help determine whether the shortcut is indeed dominant in the decision-making process, as follows:

$$\bar{p}_c - p_c > T \text{ and } p_c < \max\{p_i\}_{i=1}^{|\mathcal{C}|}, \tag{1}$$

where $x$ and $\bar{x}$ are the original and perturbed images, respectively, $p_c$ and $\bar{p}_c$ are the corresponding output probability of the classifier $f$ for class label $c$, and $T$ is set to 0.9 by default. The first prerequisite ensures that the classifier correctly categorizes a perturbed image with a high confidence score ($\bar{p}_c > T$). The second prerequisite further ensures the classifier gives a wrong prediction for the clean image $x$. With both prerequisites being satisfied, the localized shortcut should have a dominant contribution to the decision of the classifier on $\bar{x}$. Therefore, we define the area within the shortcut patch as the *dominant area*.

**Pixel Deletion Protocol**. Unknown correlations between pixels may create obstacles in estimating pixel importance. Therefore, prevailing deletion protocols posit that the assumption of independent features may not be valid at the pixel level (Alvarez Melis & Jaakkola, 2018; Chen et al., 2019). In this context, we seek to introduce the Shapley value (Shapley, 1953) to estimate pixel importance, considering the joint effect of pixels. The importance of the $i$-th pixel, denoted as $\phi_i$, is quantified by its average marginal contribution to the model's prediction across all potential combinations, expressed as follows:

$$\phi_i(v) = \frac{1}{N} \sum_{\mathcal{S} \subseteq \mathcal{N} \setminus \{i\}} \binom{N-1}{|\mathcal{S}|}^{-1} (v(\mathcal{S} \cup \{i\}) - v(\mathcal{S})), \tag{2}$$

---

**Algorithm 1:** Automatic XAI evaluation procedure

---

**Input:** A clean image dataset $\mathcal{D}$, a shortcut function $h : \mathbb{R}^N \times \mathcal{C} \mapsto \mathbb{R}^N$, an image classifier
  $f : \mathbb{R}^N \mapsto \mathbb{R}^{|\mathcal{C}|}$ trained on the perturbed version of $\mathcal{D}$, a group of XAI models
  $\mathcal{G} = \{g^{(i)}\}_{i=1}^M$ to be evaluated, and a dominance threshold $T$
**Output:** Performance metrics of $\mathcal{G}$

1   Set to gather ground-truth pixel-level explanations $\mathcal{E} \leftarrow \emptyset$
2   **for** $i \leftarrow 1$ **to** $M$ **do**
3     Set to gather $i$-th XAI model's explanations $\hat{\mathcal{E}}^{(i)} \leftarrow \emptyset$
4   **end**
5   **for** $(\boldsymbol{x}, c) \in \mathcal{D}$ **do**
6     Obtain the perturbed image $\bar{\boldsymbol{x}} \leftarrow h(\boldsymbol{x}, c)$
7     $\bar{\boldsymbol{p}} \leftarrow f(\bar{\boldsymbol{x}})$
8     $\boldsymbol{p} \leftarrow f(\boldsymbol{x})$
9     **if** $\bar{p}_c - p_c > T$ and $p_c < \max\{p_i\}_{i=1}^{|\mathcal{C}|}$ **then**
10       Assign zero importance to pixels outside the *dominant area* $\phi_{\text{out}} \leftarrow \boldsymbol{0}$
11       Estimate Shapley value $\phi_{\text{in}}$ (Equation 2)
12       Obtain the ground-truth pixel attributions $\phi \leftarrow \phi_{\text{in}} \bigcup \phi_{\text{out}}$
13       $\mathcal{E} \leftarrow \mathcal{E} \bigcup \phi$
14       **for** $i \leftarrow 1$ **to** $M$ **do**
15         Predict the pixel attributions using the $i$-th XAI method $\hat{\mathcal{E}}^{(i)} \leftarrow \hat{\mathcal{E}}^{(i)} \bigcup g^{(i)}(f, \bar{\boldsymbol{x}}, c)$
16       **end**
17     **end**
18   **end**
19   **for** $i \leftarrow 1$ **to** $M$ **do**
20     Compute the quantitative metrics $\text{M}(\hat{\mathcal{E}}^{(i)}, \mathcal{E})$
21   **end**

---

where $N$ represents the total number of pixels, $\mathcal{N}$ is the universal set of pixels and $\mathcal{S}$ is a coalition of pixels. The value function $v(\mathcal{S})$ is defined as $p'_c$, where $\boldsymbol{p}' = f(\bar{\boldsymbol{x}}_s)$, and $\bar{\boldsymbol{x}}_s$ is a variant of $\bar{\boldsymbol{x}}$ with the $i$-th pixel deleted when $i \notin \mathcal{S}$.

Upon fulfilling the prerequisites for the *dominant area*, we assume pixels external to this area have minimal contribution to the decision-making process, analogous to *Null Players* in a cooperative game (Shapley, 1953), and assign an importance score of zero to these pixels. This assumption is valuable as null players are excluded from Shapley value computation due to their non-contributory nature. As such, it helps simplify computations without compromising equity (Shapley, 1953). Based on this assumption, the computation of the Shapley value can be confined to pixels in the *dominant area*. Given the dimensional constraints of the image classifier's input, a pixel cannot be truly deleted, and a placeholder is necessary. We choose the original value of the clean image as the placeholder for shortcut pixels inside the *dominant area* rather than a trivial constant (*e.g.*, 0) to avoid creating unintended decision rules and causing unexpected classifier behaviors. The procedure for automatic evaluation is delineated in Algorithm 1.

In addition, it is intractable to compute the exact Shapley value, as the computation of the Shapley value becomes exponentially complex along with increasing the number of pixels. Instead, we follow Mitchell et al. (2022) to substitute the exact Shapley value with the average of Monte Carlo sampling estimates over 5 trials. We define the estimated pixel importance as the pseudo ground truth for further evaluation. The verification of our constituted ground truth's correctness is detailed in Sec. 4.3.

### 3.3 METRICS

We first measure the ability of XAI models to detect important areas at a coarse level. We define an image with/without a *dominant area* as a positive/negative sample. For a positive sample, if the most important pixels ranked by an XAI model are located in the *dominant area*, the method is identified as yielding a correct prediction. The true positive rate is then defined as hit accuracy (HA),

Table 1: Test accuracy (%) and dominant rate (%) of classifiers trained on the perturbed datasets.

| Model | CIFAR-10 | | | CIFAR-100 | | | ImageNet | | |
|---|---|---|---|---|---|---|---|---|---|
| | C-Set | P-Set | DR | C-Set | P-Set | DR | C-Set | P-Set | DR |
| VGG | 25.56 | 97.60 | 58.50 | 22.56 | 93.65 | 59.47 | 36.12 | 85.19 | 30.20 |
| ResNet | 23.72 | 99.65 | 56.39 | 28.36 | 94.12 | 61.34 | 34.11 | 89.10 | 37.50 |
| ViT | 37.52 | 98.17 | 53.73 | 26.19 | 87.85 | 48.52 | 38.85 | 77.56 | 27.05 |
| SwinT | 32.41 | 98.97 | 55.96 | 27.36 | 91.69 | 50.62 | 35.34 | 80.75 | 29.90 |

Table 2: Test accuracy (%) of classifiers trained on the original datasets.

| Model | CIFAR-10 | | CIFAR-100 | | ImageNet | |
|---|---|---|---|---|---|---|
| | C-Set | P-Set | C-Set | P-Set | C-Set | P-Set |
| VGG | 93.21 | 92.73 | 74.86 | 74.25 | 72.84 | 72.80 |
| ResNet | 94.57 | 94.12 | 76.22 | 75.67 | 75.49 | 75.49 |
| ViT | 93.35 | 93.01 | 73.92 | 73.35 | 74.80 | 74.78 |
| SwinT | 94.02 | 93.56 | 77.32 | 76.79 | 80.96 | 80.93 |

inspired by Zhang et al. (2018). To evaluate the pixel-level importance predicted by XAI models, we calculate the Intersection over Union (IoU) between the top $k$ pixels identified by the XAI model and the ground truth, respectively, across multiple $k$ values. By assigning different weights $w_k$ to each $k$ value, we define the weighted average IoU (WIoU) as follows:

$$\text{WIoU} = \sum_k w_k \cdot \frac{|\text{top-}k(g^{(i)}(f, \bar{\boldsymbol{x}}, c)) \cap \text{top-}k(\boldsymbol{\phi})|}{|\text{top-}k(g^{(i)}(f, \bar{\boldsymbol{x}}, c)) \cup \text{top-}k(\boldsymbol{\phi})|}, \tag{3}$$

which reveals the accuracy of the XAI model in identifying key pixels that influence model predictions. Assigning higher weights to smaller $k$ values can emphasize the superiority of correctly identifying the pixels with higher importance. Notably, the WIoU ranges from 0 to 1, with higher values indicating better model performance and alignment with the ground truth.

## 4 EXPERIMENTS

### 4.1 IMPLEMENTATION DETAILS

The evaluation is conducted on the perturbed CIFAR-10, CIFAR-100, and ImageNet (Russakovsky et al., 2015) datasets. $K$ and $D$ are set to 25, 25 for CIFAR-10 and CIFAR-100, while 225 and 64 for ImageNet. $\alpha$ is set to 0.1 for CIFAR-10, 0.3 for CIFAR-100 and 0.5 for ImageNet. Notably, a perturbation location is shared by multiple categories (*e.g.*, 5 categories for ImageNet) but convolutional kernels are unique for each category on CIFAR-100 and ImageNet.

We implement four DNN architectures, *i.e.*, VGG (Simonyan & Zisserman, 2014), ResNet (He et al., 2016), Vision Transformer (ViT) (Dosovitskiy et al., 2020) and Swin Transformer (SwinT) (Liu et al., 2021) on the three perturbed datasets. On CIFAR-10 and CIFAR-100 datasets, we use VGG-16, ResNet-18, customized Vision Transformer (ViT-C) and customized Swin Transformer (SwinT-C). On ImageNet dataset, we use standard VGG-16, ResNet-50, ViT-B and SwinT-B models. The customized structure parameters and detailed training settings are shown in the supplementary material.

We implement fourteen saliency-based XAI models, categorized into gradient-based, perturbation-based, and attention-based approaches. Gradient-based methods include: Deconvnet (Zeiler & Fergus, 2014), Gradient (Simonyan et al., 2014), Guided Backpropagation (GBP) (Springenberg et al., 2015), Input × Gradient (IxG) (Shrikumar et al., 2016), Grad-CAM (Selvaraju et al., 2017a), Integrated Gradients (IG) (Sundararajan et al., 2017), DeepLift (Shrikumar et al., 2017) and GradientSHAP (Lundberg & Lee, 2017). Perturbation-based methods include: Occlusion (Zeiler & Fergus, 2014), LIME (Ribeiro et al., 2016), KernelSHAP (Lundberg & Lee, 2017), RISE (Petsiuk et al., 2018) and Extremal Perturbation (EP) (Fong et al., 2019). Deconvnet is not applicable to Transformers due to its reliance on deconvolution. Instead, for Vision Transformers and Swin Transformers, we utilize attention-based methods such as Attention Rollout (AR) (Abnar & Zuidema, 2020).

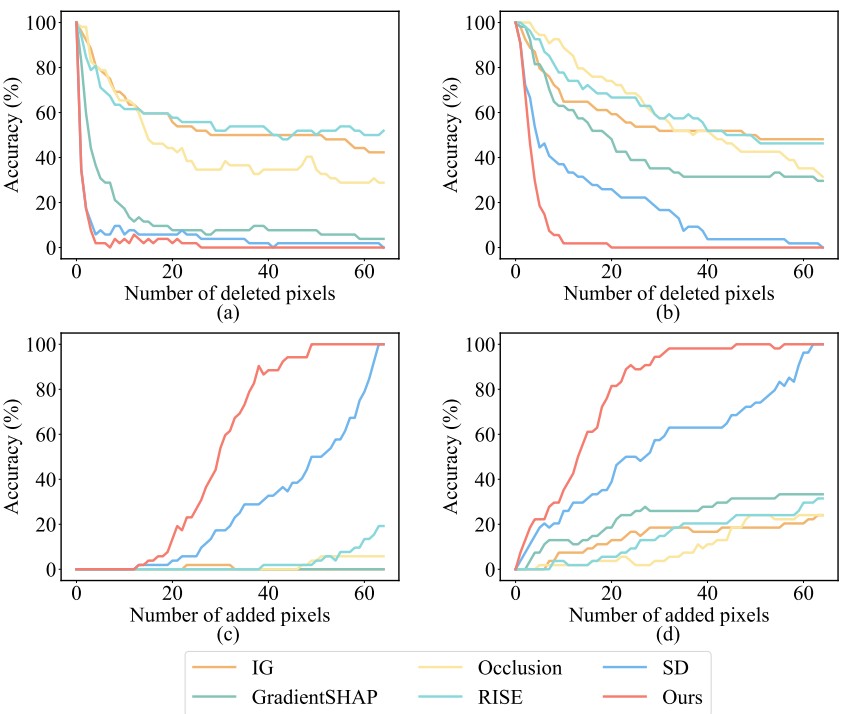

Figure 2: Ablation on various XAI methods assessed on ImageNet datasets using ResNet/ViT after deleting/adding top-64 pixels. Specifically, we have (a) Deletion curve (ImageNet + ResNet-50), (b) Deletion curve (ImageNet + ViT-B), (c) Addition curve (ImageNet + ResNet-50), and (d) Addition curve (ImageNet + ViT-B).

Table 3: Normalized area under curves (AUC) of various deletion/addition curves on CIFAR-10/CIFAR-100/ImageNet using ResNet/ViT after deleting/adding top-$D$ pixels.

| Method | C10+ResNet | | C10+ViT | | C100+ResNet | | C100+ViT | | IN+ResNet | | IN+ViT | |
|---|---|---|---|---|---|---|---|---|---|---|---|---|
| | Del | Add | Del | Add | Del | Add | Del | Add | Del | Add | Del | Add |
| **Gradient-based** | | | | | | | | | | | | |
| Deconvnet | 0.759 | 0.152 | - | - | 0.758 | 0.042 | - | - | 1.000 | 0.015 | - | - |
| Gradient | 0.835 | 0.120 | 0.693 | 0.356 | 0.792 | 0.032 | 0.573 | 0.158 | 0.884 | 0.000 | 0.617 | 0.169 |
| GBP | 0.759 | 0.152 | 0.626 | 0.425 | 0.758 | 0.042 | 0.525 | 0.144 | 0.171 | 0.001 | 0.576 | 0.163 |
| IxG | 0.880 | 0.084 | 0.865 | 0.187 | 0.869 | 0.017 | 0.660 | 0.087 | 0.907 | 0.000 | 0.795 | 0.007 |
| Grad-CAM | 0.880 | 0.174 | 0.932 | 0.045 | 0.926 | 0.012 | 0.956 | 0.095 | 0.992 | 0.000 | 0.829 | 0.004 |
| IG | 0.816 | 0.209 | 0.751 | 0.409 | 0.609 | 0.060 | 0.540 | 0.212 | 0.560 | 0.003 | 0.581 | 0.143 |
| DeepLift | 0.880 | 0.084 | 0.865 | 0.187 | 0.869 | 0.017 | 0.660 | 0.087 | 0.907 | 0.000 | 0.795 | 0.007 |
| GradientSHAP | 0.700 | 0.181 | 0.620 | 0.433 | 0.327 | 0.041 | 0.415 | 0.203 | 0.129 | 0.000 | 0.444 | 0.233 |
| **Perturbation-based** | | | | | | | | | | | | |
| Occlusion | 0.770 | 0.276 | 0.697 | 0.367 | 0.709 | 0.105 | 0.573 | 0.320 | 0.445 | 0.014 | 0.616 | 0.100 |
| LIME | 1.000 | 0.002 | 0.920 | 0.166 | 0.976 | 0.008 | 0.958 | 0.030 | 1.000 | 0.000 | 1.000 | 0.000 |
| KernelSHAP | 0.961 | 0.019 | 0.948 | 0.093 | 0.941 | 0.006 | 0.813 | 0.022 | 1.000 | 0.000 | 1.000 | 0.000 |
| RISE | 0.846 | 0.194 | 0.796 | 0.276 | 0.580 | 0.116 | 0.633 | 0.272 | 0.574 | 0.025 | 0.623 | 0.144 |
| EP | 0.882 | 0.110 | 0.909 | 0.236 | 0.856 | 0.031 | 0.937 | 0.035 | 0.950 | 0.000 | 0.988 | 0.000 |
| **Attention-based** | | | | | | | | | | | | |
| AR | - | - | 1.000 | 0.044 | - | - | 0.877 | 0.080 | - | - | 0.722 | 0.117 |
| SD | 0.275 | 0.733 | 0.167 | 0.885 | 0.133 | 0.478 | 0.215 | 0.589 | 0.054 | 0.270 | 0.197 | 0.543 |
| Ours | **0.139** | **0.880** | **0.162** | **0.893** | **0.115** | **0.515** | **0.159** | **0.596** | **0.026** | **0.532** | **0.056** | **0.787** |

## 4.2 CLASSIFIER PERFORMANCE

We test the classifiers using two different test settings. The clean test set (C-Set) contains original clean images. The perturbed test set (P-Set) is perturbed in the same way as the training set. The test accuracy is listed in Table 1. The performance gap between C-Set and P-Set proves the classifier adopts the shortcut as a decision rule preferentially. We also compute the dominant rate (DR), *i.e.*, the rate of the images satisfying dominant prerequisites, and the results are listed in the same table. Additionally, we test classifiers trained on the original clean dataset on C-Set and P-Set, the

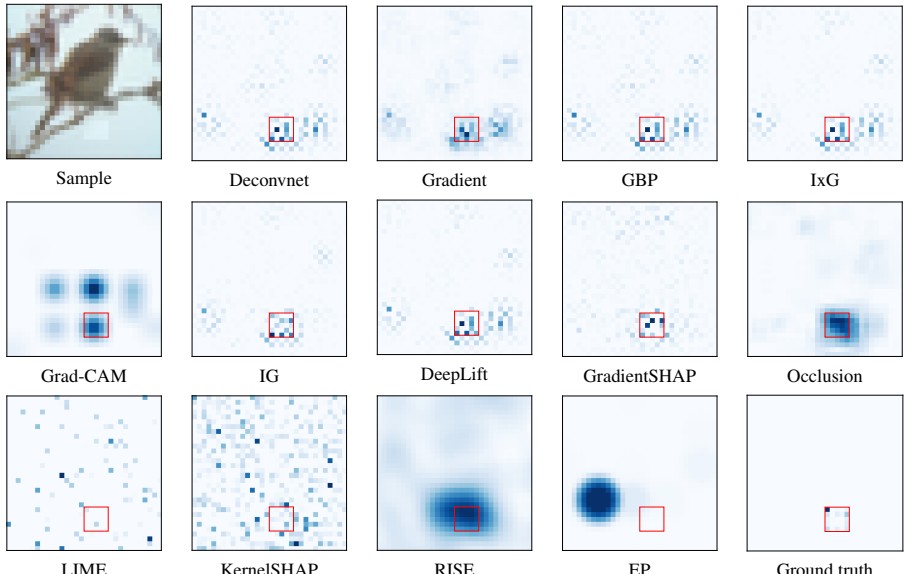

| Sample | Deconvnet | Gradient | GBP | IxG |

| Grad-CAM | IG | DeepLift | GradientSHAP | Occlusion |

| LIME | KernelSHAP | RISE | EP | Ground truth |

Figure 3: Close-ups of qualitative examples in CIFAR-10 showcasing various explanations for ResNet-18. The red circle-enclosed areas represent the *dominant areas* associated with the ground-truth category. Each unit patch corresponds to a single pixel in the original image. The color saturation of the patch indicates the degree of the pixel's contribution to the correct prediction, with more saturated colors indicating a greater contribution.

consistent performance (see Table 2) confirms the local perturbation has a minimal impact on the decision rules in the original dataset.

### 4.3 Verification of Ground Truth

To validate the correctness and reliability of our proposed ground truth, we start by deleting top-$D$ pixels one by one based on the descending ranks provided by our proposed ground truth and the evaluated XAI methods. With this process, we observe variations in prediction accuracy. We also add a single deletion (SD) baseline for comparison, which estimates pixel importance in the *dominant area* by single deletion protocol. The results of deleting top-64 pixels on ImageNet can be found in Figure 2 (a,b), indicating that our method exhibits the fastest descending trend, and the prediction accuracy drops to near zero with only 10 pixels being deleted. Additionally, We add top-$D$ shortcut pixels one by one on clean images in descending orders. Our proposed method consistently exhibits the fastest ascending trend, as shown in Figure 2 (c,d). We further calculate the AUC presented in Table 3, noting that for deletion curves, a smaller area is preferable; whereas for addition curves, a larger area is indicative of better performance. The observation of our method outperforming SD implies that the feature independence assumption is flawed, resulting in sub-optimal performance. In addition, our method shows superior performance over KernelSHAP applied to the entire images. This can be attributed to the KernelSHAP's dependency on sample size, and when dealing with a large feature set, like exceeding a thousand, collecting a sufficient number of samples to compute Shapley value becomes prohibitively time-consuming and even intractable. Our proposed method achieves the best performance on all dataset-DNN combinations, and all these findings demonstrate that our method offers a more accurate explanation than existing XAI methods, therefore we adopt it as the pseudo ground truth for evaluation. More results can be found in the supplementary material.

### 4.4 Evaluation Results and Analysis

**Comparison among XAI Methods**. Qualitative illustrations of CIFAR-10 are depicted in Figure 3. Although multiple methods focus on the shortcut area, the pixel attributions exhibit variations. The quantitative results for ResNet and ViT are presented in Table 4, with additional results available in the supplementary material. When computing WIoU, the specific $k$ values and their corresponding weights $w_k$ are defined as follows: $k = \{25, 20, 15, 10, 5, 3, 1\}$ and $w_k = \{1, 3, 5, 10, 15, 20, 25\}$.

Table 4: Evaluation results of various XAI methods conducted on CIFAR-10/CIFAR-100/ImageNet datasets using ResNet/ViT. Metrics: HA (%) and WIoU.

| Method | C10+ResNet | | C10+ViT | | C100+ResNet | | C100+ViT | | IN+ResNet | | IN+ViT | |
|---|---|---|---|---|---|---|---|---|---|---|---|---|
| | HA | WIoU | HA | WIoU | HA | WIoU | HA | WIoU | HA | WIoU | HA | WIoU |
| **Gradient-based** | | | | | | | | | | | | |
| Deconvnet | 33.27 | 0.056 | - | - | 36.75 | 0.025 | - | - | 1.92 | 0.000 | - | - |
| Gradient | 32.38 | 0.048 | 38.55 | 0.095 | 38.37 | 0.031 | **70.00** | 0.090 | 9.62 | 0.006 | 35.19 | 0.112 |
| GBP | 33.27 | 0.056 | 44.91 | 0.117 | 36.75 | 0.022 | 69.40 | 0.082 | 69.23 | 0.043 | 38.89 | **0.150** |
| IxG | 24.69 | 0.031 | 24.91 | 0.046 | 27.15 | 0.017 | 48.20 | 0.051 | 3.85 | 0.001 | 14.81 | 0.030 |
| Grad-CAM | **59.93** | 0.060 | 11.32 | 0.012 | 21.79 | 0.027 | 14.67 | 0.019 | 0.00 | 0.000 | 11.89 | 0.018 |
| IG | 43.47 | 0.076 | 48.91 | **0.173** | 60.49 | **0.076** | 62.40 | **0.142** | 34.62 | 0.020 | 37.04 | 0.108 |
| DeepLift | 24.69 | 0.032 | 24.91 | 0.045 | 27.15 | 0.016 | 48.20 | 0.053 | 3.85 | 0.001 | 14.81 | 0.030 |
| GradientSHAP | **63.51** | **0.087** | **63.82** | **0.188** | **91.38** | **0.107** | **90.00** | **0.125** | **94.23** | **0.107** | **53.70** | **0.118** |
| **Perturbation-based** | | | | | | | | | | | | |
| Occlusion | 50.27 | **0.089** | 47.45 | 0.076 | 53.50 | 0.073 | 42.20 | 0.087 | **82.69** | 0.016 | **79.63** | 0.027 |
| LIME | 4.29 | 0.007 | 23.27 | 0.017 | 7.80 | 0.003 | 21.40 | 0.016 | 0.00 | 0.000 | 1.85 | 0.002 |
| KernelSHAP | 9.30 | 0.009 | 10.36 | 0.010 | 17.24 | 0.009 | 19.00 | 0.020 | 3.85 | 0.002 | 1.85 | 0.000 |
| RISE | 41.68 | 0.069 | **52.18** | 0.064 | **68.78** | 0.056 | 44.20 | 0.060 | 44.23 | **0.048** | 37.04 | 0.040 |
| EP | 34.35 | 0.053 | 36.36 | 0.048 | 27.48 | 0.036 | 42.00 | 0.032 | 17.31 | 0.002 | 12.96 | 0.003 |
| **Attention-based** | | | | | | | | | | | | |
| AR | - | - | 17.31 | 0.055 | - | - | 30.77 | 0.025 | - | - | 32.21 | 0.071 |

The HA results indicate that most XAI methods struggle to accurately identify the shortcut in over half of the test samples, even at a coarse level. The WIoU results suggest that all methods exhibit limited capability in offering precise explanations for individual pixels. Both results highlight the demand for explanation models with higher correctness, which is in line with the findings in Hesse et al. (2024).

Most gradient-based methods attribute importance scores directly to individual pixels, with the exception of Grad-CAM, which may exhibit reduced sensitivity to fine-grained pixel-level variations due to its reliance on downsampled feature maps. Comparatively, Gradient and IG show better performance and more precise explanations. Nonetheless, Gradient can be susceptible to gradient vanishing issues. IG addresses the gradient vanishing issues by integrating gradients along a single baseline path, although the complex interactions between pixels may not be faithfully captured. GradientSHAP demonstrates the best performance across multiple dataset-DNN combinations. This approach leverages the Shapley value framework to deliver a robust attribution. However, its stochastic sampling nature and the assumption of feature independence may introduce variability at the pixel level.

Perturbation-based methods, such as Occlusion and RISE, tend to attribute proximity significance to the recognized important area. While proficient at capturing the importance of regions, these methods may struggle with precise pixel-level ranking due to their coarse masking strategies. Both LIME and KernelSHAP generally exhibit subpar performance, underscoring their limited effectiveness in high-dimensional and intricate pixel space. Additionally, the performance of AR is notably modest across all datasets. During the aggregation of attention maps, detailed pixel-specific information may be compromised, leading to high-level features that are often too generalized to accurately portray the importance of individual pixels.

**Comparison with Related Protocols**. We compare our protocol with four well-established protocols: the incremental-deletion score (IDS) (Samek et al., 2016), the OOD single-deletion score (SDS) (Selvaraju et al., 2017b), the single deletion protocol in FunnyBirds (Hesse et al., 2023), and the in-domain single-deletion score (IDSDS) (Hesse et al., 2024). Specifically, we compare the rankings of Gradient, IxG, Grad-CAM, IG (with an all-zero baseline), and RISE given by different protocols. The evaluation of our protocol is conducted in ImageNet with ResNet-50, ranking the XAI methods according to their WIoU. We follow the settings in Hesse et al. (2024) to obtain the rankings from other protocols with ResNet-50. Detailed ranking results are available in Table 5.

Among these methods, Grad-CAM performs best for IDS, SDS, and IDSDS. However, it performs worst for our protocol. The limitation of these three deletion protocols lies in their loss of granularity, as they evaluate at a patch level and are unable to assess pixel importance within each patch (Hesse et al., 2024). On the other, Grad-CAM exhibits lower resolution than competing methods. Therefore, Grad-CAM may perform well when evaluated at a patch level but fails at a pixel level. Interestingly, when IG switches from an all-zero baseline to a uniform baseline, its performance improves for IDSDS, contrary to the result in Funnybirds. The inconsistency suggests that using a single base-

Table 5: XAI methods ranking by different evaluation protocols.

| Methods | IDS | SDS | FunnyBirds | IDSDS | Ours |
|---------|-----|-----|------------|-------|------|
| Gradient | 3 | 4 | 4 | 4 | 3 |
| IxG | 5 | 5 | 5 | 5 | 4 |
| Grad-CAM | 1 | 1 | 3 | 1 | 5 |
| IG | 4 | 2 | 1 | 2 | 2 |
| RISE | 2 | 3 | 2 | 3 | 1 |

line for integrating gradients may cause variable performance, and evaluation protocols might be biased toward different baselines. Incorporating multiple baselines could reduce evaluation bias. Additionally, GradientSHAP, an extension of IG that computes the expectations of gradients over multiple baselines, demonstrates exceptional performance under our protocol. Thus, the explanation correctness may also improve by incorporating multiple baselines.

## 5 CONCLUSION

We have presented a novel method for evaluating saliency-based explainable artificial intelligence (XAI) models at the raw image pixel level, focusing on shortcuts. To achieve this, we propose a protocol, which involves training DNN-based classifiers on datasets with localized shortcuts. This innovative approach allows for generating pixel-level ground-truth explanations using Shapley value estimation, and further evaluating saliency-based XAI models. Our experimental findings reveal the limited capacity of current saliency-based XAI methods in explaining model behaviors, highlighting the demand to prioritize explanations of the model's authentic learning process at a granular pixel level. Developing a universal protocol for evaluating explanations in real-world scenarios is an interesting venue for future work.

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
