# Supplementary Material

## A  Implemetation Details

On CIFAR-10 and CIFAR-100 datasets, we use VGG-16, ResNet-18, customized Vision Transformer (ViT-C) and customized Swin Transformer (SwinT-C). Considering the small image size, we modify the ResNet-18 model to address the potential loss of information caused by the initial down-sampling convolution with the kernel size 7 and max pooling operation. Specifically, we replaced this step with a convolution kernel (size 3) to preserve more useful information. For the ViT-C and SwinT-C models, we customized the parameters to account for the same concern. The customized parameters are shown in Table 1. On ImageNet dataset, we use standard VGG-16, ResNet-50, ViT-B and SwinT-B models. The training hyperparameters of different dataset-DNN combinations are listed in Table 2, where, $l_0$ is the initial learning rate, and Warm-up refers to the number of iterations for linear warm-up. The learning rate follows cosine annealing.

Table 1: Customized architecture parameters of ViT and SwinT.

| Model | Depth | Patch-size | Token Dimension | Heads | MLP-ratio | Window-size |
|-------|-------|-----------|-----------------|-------|-----------|-------------|
| ViT-C | 9 | 4 | 192 | 12 | 2 | - |
| SwinT-C | [2,6,4] | 2 | 96 | [3,6,12] | 2 | 4 |

Table 2: Training hyperparameters.

| Model | Optimizer | Batchsize | $l_0$ | Epoch | Warm-up |
|-------|-----------|-----------|-------|-------|---------|
| **CIFAR-10** | | | | | |
| VGG-16 | SGD | 100 | 0.1 | 50 | - |
| ResNet-18 | Adam | 100 | 0.001 | 50 | - |
| ViT-C | AdamW | 100 | 0.001 | 100 | 7500 |
| SwinT-C | AdamW | 100 | 0.001 | 100 | 5000 |
| **CIFAR-100** | | | | | |
| VGG-16 | SGD | 100 | 0.1 | 50 | - |
| ResNet-18 | SGD | 100 | 0.1 | 50 | - |
| ViT-C | AdamW | 100 | 0.001 | 100 | 7500 |
| SwinT-C | AdamW | 100 | 0.001 | 100 | 7500 |
| **ImageNet** | | | | | |
| VGG-16 | SGD | 256 | 0.1 | 100 | 1000 |
| ResNet-50 | SGD | 256 | 0.1 | 100 | 1000 |
| ViT-B | AdamW | 256 | 0.0001 | 100 | 1000 |
| SwinT-B | AdamW | 256 | 0.0001 | 100 | 1000 |

# B  VERIFICATION OF GROUND TRUTH

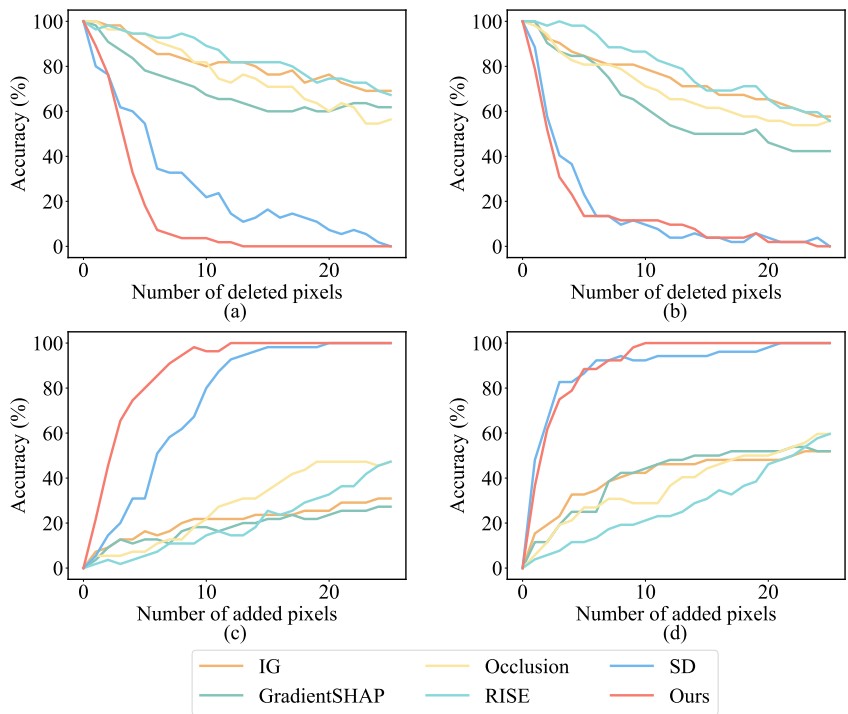

Figure 1: Ablation curves on various XAI methods assessed on CIFAR-10 datasets using ResNet/ViT after deleting/adding top-25 pixels. Specifically, we have (a) Deletion curve (CIFAR-10 + ResNet-18), (b) Deletion curve (CIFAR-10 + ViT-C), (c) Addition curve (CIFAR-10 + ResNet-18), and (d) Addition curve (CIFAR-10 + ViT-C).

Table 3: Normalized Area under curves (AUC) of different deletion/addition curves on CIFAR-10/CIFAR-100/ImageNet using VGG/SwinT after deleting/adding top-$D$ pixels.

| Method | C10+VGG | | C10+SwinT | | C100+VGG | | C100+SwinT | | IN+VGG | | IN+SwinT | |
|---|---|---|---|---|---|---|---|---|---|---|---|---|
| | Del | Add | Del | Add | Del | Add | Del | Add | Del | Add | Del | Add |
| **Gradient-based** | | | | | | | | | | | | |
| Deconvnet | 0.602 | 0.292 | - | - | 0.633 | 0.198 | - | - | 0.530 | 0.007 | - | - |
| Gradient | 0.635 | 0.234 | 0.836 | 0.154 | 0.851 | 0.098 | 0.432 | 0.264 | 0.798 | 0.019 | 0.423 | 0.170 |
| GBP | 0.599 | 0.251 | 0.821 | 0.167 | 0.612 | 0.201 | 0.456 | 0.176 | 0.341 | 0.010 | 0.501 | 0.132 |
| IxG | 0.822 | 0.156 | 0.885 | 0.102 | 0.872 | 0.098 | 0.576 | 0.063 | 0.960 | 0.002 | 0.768 | 0.053 |
| Grad-CAM | 0.899 | 0.208 | 0.948 | 0.096 | 0.901 | 0.076 | 0.925 | 0.035 | 0.962 | 0.004 | 0.859 | 0.002 |
| IG | 0.708 | 0.256 | 0.634 | 0.209 | 0.686 | 0.109 | 0.577 | 0.134 | 0.573 | 0.008 | 0.532 | 0.160 |
| DeepLift | 0.822 | 0.156 | 0.885 | 0.102 | 0.872 | 0.098 | 0.576 | 0.063 | 0.960 | 0.002 | 0.768 | 0.053 |
| GradientSHAP | 0.534 | 0.277 | 0.536 | 0.268 | 0.532 | 0.193 | 0.341 | 0.209 | 0.198 | 0.007 | 0.396 | 0.224 |
| **Perturbation-based** | | | | | | | | | | | | |
| Occlusion | 0.682 | 0.293 | 0.586 | 0.287 | 0.687 | 0.132 | 0.405 | 0.431 | 0.472 | 0.009 | 0.610 | 0.114 |
| LIME | 0.978 | 0.011 | 0.934 | 0.107 | 0.955 | 0.057 | 0.864 | 0.097 | 0.995 | 0.001 | 0.999 | 0.001 |
| KernelSHAP | 0.954 | 0.018 | 0.885 | 0.099 | 0.935 | 0.046 | 0.856 | 0.019 | 0.979 | 0.000 | 0.998 | 0.003 |
| RISE | 0.716 | 0.205 | 0.796 | 0.172 | 0.789 | 0.102 | 0.761 | 0.157 | 0.561 | 0.008 | 0.534 | 0.127 |
| EP | 0.891 | 0.099 | 0.899 | 0.187 | 0.811 | 0.092 | 0.768 | 0.103 | 0.967 | 0.003 | 0.999 | 0.000 |
| **Attention-based** | | | | | | | | | | | | |
| AR | - | - | 0.967 | 0.035 | - | - | 0.836 | 0.112 | - | - | 0.781 | 0.121 |
| SD | 0.262 | 0.789 | 0.217 | 0.823 | 0.124 | 0.571 | 0.244 | 0.414 | 0.063 | 0.417 | 0.175 | 0.516 |
| Ours | **0.128** | **0.867** | **0.177** | **0.901** | **0.118** | **0.634** | **0.135** | **0.511** | **0.021** | **0.636** | **0.078** | **0.685** |

## C EVALUATION RESULTS

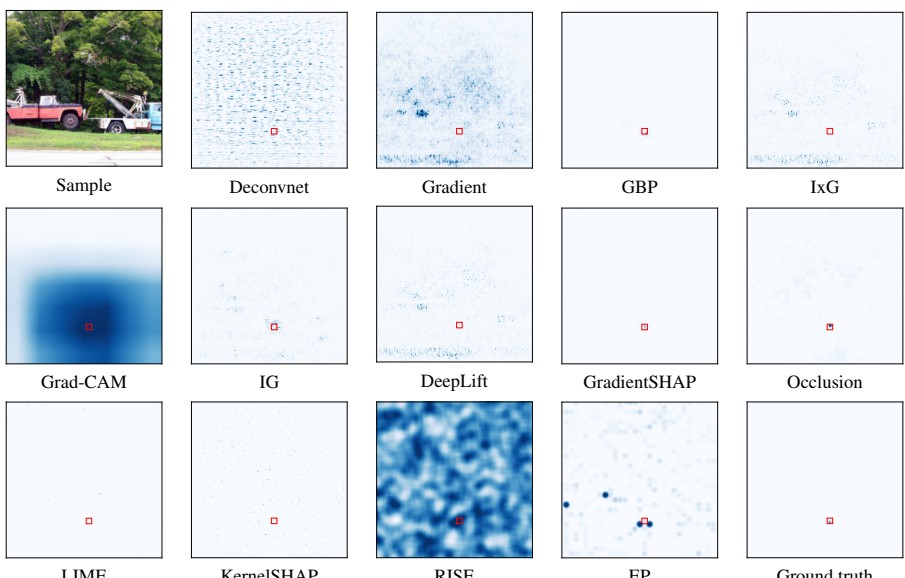

Figure 2: Close-ups of qualitative examples in ImageNet showcasing various explanations for ResNet-50. The red circle-enclosed areas represent the *dominant areas* associated with the ground-truth category. The color saturation of the patch indicates the degree of the pixel's contribution to the correct prediction, with more saturated colors indicating a greater contribution.

Table 4: Evaluation results of different XAI methods conducted on CIFAR-10/CIFAR-100/ImageNet using VGG/SwinT.

| Method | C10+VGG | | C10+SwinT | | C100+VGG | | C100+SwinT | | IN+VGG | | IN+SwinT | |
|---|---|---|---|---|---|---|---|---|---|---|---|---|
| | HA | WIoU | HA | WIoU | HA | WIoU | HA | WIoU | HA | WIoU | HA | WIoU |
| **Gradient-based** | | | | | | | | | | | | |
| Deconvnet | **81.25** | **0.153** | - | - | **78.18** | **0.124** | - | - | 29.67 | 0.020 | - | - |
| Gradient | 74.49 | 0.121 | 27.59 | 0.025 | 32.82 | 0.029 | 48.39 | 0.051 | 4.00 | 0.000 | 35.93 | 0.104 |
| GBP | 81.08 | 0.117 | 32.59 | 0.031 | 70.27 | **0.131** | 46.98 | 0.063 | 53.67 | 0.056 | 33.90 | 0.088 |
| IxG | 45.44 | 0.023 | 17.59 | 0.010 | 27.32 | 0.011 | 33.06 | 0.030 | 5.00 | 0.005 | 16.27 | 0.029 |
| Grad-CAM | 13.18 | 0.009 | 7.98 | 0.005 | 5.67 | 0.008 | 8.14 | 0.014 | 1.33 | 0.001 | 10.17 | 0.007 |
| IG | 46.45 | 0.096 | 40.69 | 0.098 | 52.58 | 0.083 | 53.63 | 0.105 | 27.00 | 0.015 | 34.92 | **0.116** |
| DeepLift | 45.44 | 0.023 | 17.59 | 0.010 | 27.32 | 0.011 | 33.06 | 0.030 | 5.00 | 0.005 | 16.27 | 0.029 |
| GradientSHAP | 72.30 | **0.132** | **64.31** | **0.117** | **88.32** | 0.122 | **90.12** | **0.162** | **89.33** | **0.100** | **50.85** | **0.135** |
| **Perturbation-based** | | | | | | | | | | | | |
| Occlusion | 70.14 | 0.095 | **46.67** | **0.102** | 62.50 | 0.073 | **70.00** | **0.106** | **72.33** | **0.079** | 45.76 | 0.011 |
| LIME | 3.04 | 0.002 | 17.78 | 0.022 | 1.79 | 0.003 | 8.00 | 0.012 | 4.00 | 0.000 | 5.08 | 0.001 |
| KernelSHAP | 22.97 | 0.005 | 6.12 | 0.007 | 16.07 | 0.009 | 10.00 | 0.021 | 2.00 | 0.005 | 5.76 | 0.002 |
| RISE | 58.61 | 0.073 | 35.56 | 0.061 | 39.29 | 0.056 | 32.00 | 0.034 | 35.33 | 0.036 | 33.22 | 0.056 |
| EP | 36.25 | 0.034 | 26.67 | 0.031 | 32.14 | 0.036 | 32.00 | 0.036 | 10.00 | 0.001 | 10.85 | 0.002 |
| **Attention-based** | | | | | | | | | | | | |
| AR | - | - | 12.14 | 0.014 | - | - | 27.86 | 0.025 | - | - | 34.58 | 0.075 |