# OpenReview forum: "Decision Rules are in the Pixels: Towards Pixel-level Evaluation of Saliency-based XAI Models"
_ICLR.cc/2025/Conference — ICLR 2025 Conference Withdrawn Submission_

### Official Review · Reviewer_Vhj6 · 2024-10-29

**Soundness:** 3
**Presentation:** 2
**Contribution:** 1
**Rating:** 3
**Confidence:** 4

**Summary:**

This paper proposes a method to provide ground truth for XAI saliency map methods, which attempt to measure how important different pixels (or segments) in an image are to a model's classification decision and visualise the importance as a saliency map. The proposed method adds a class-specific watermark to each image. The main distinction between this paper and previous ones seems to be that the watermark is more complex, so individual pixels in the watermark are not equally important. Experiments with various XAI saliency map methods on image datasets replicate previous work showing that saliency map output tends to be a very poor reflection of the ground truth importance of pixels in a classification task.

**Strengths:**

- This work evaluates XAI saliency methods against a known ground truth, which is an important evaluation that is often ignored in XAI
- The results provide a nice confirmation of previous work showing that most XAI saliency methods do not work as expected when tested with known ground truth; they also seem to confirm previous work (https://proceedings.neurips.cc/paper_files/paper/2018/file/294a8ed24b1ad22ec2e7efea049b8737-Paper.pdf) showing that GradCAM comes closest to having the expected behavior

**Weaknesses:**

- This work does not seem very novel, since the main departure from previous work seems to be a slightly different watermarking process that makes it more feasible to rank the importance of individual pixels within the watermark. It's not clear to me how important that change is for evaluating saliency maps, given that overall this work confirms and replicates existing findings.
- Although ground truth evaluation of XAI methods is limited, there has been some nice work extensively evaluating saliency maps (e.g., https://pmc.ncbi.nlm.nih.gov/articles/PMC10558518/, https://proceedings.neurips.cc/paper_files/paper/2018/file/294a8ed24b1ad22ec2e7efea049b8737-Paper.pdf). This work is not discussed, so it's unclear how the current paper is extending the existing work.
- The novel watermarking process may have some downsides. In particular, the watermark location is also class-specific (but not unique to a class, because there are more classes than possible locations), which I don't think I've seen in previous work. This makes the saliency map output slightly harder to interpret, because the LACK of a watermark in a candidate location could also be an important feature for recognition (which might be what the Fig 3 GradCAM result shows).

**Questions:**

- What is the reasoning behind having the watermark location be class-specific, in addition to having a class-specific pattern? Generally we assume the "important" features for an image can appear anywhere, so shouldn't the watermark location be random within each class? (This would also help ensure the models don't learn "lack of watermark in location x,y" as a potentially-important feature for decision making.)

---

### Official Review · Reviewer_dbQ6 · 2024-11-02

**Soundness:** 2
**Presentation:** 3
**Contribution:** 2
**Rating:** 5
**Confidence:** 4

**Summary:**

The manuscript addresses the assessment of saliency-based interpretable models at the pixel level, aiming to verify the accuracy of explainable AI (XAI) methods. The proposed evaluation protocol introduces a shortcut decision rule with three beneficial properties: dominance as the decision rule, minimal impact on existing patterns, and spatial localization. The effectiveness of this evaluation method is substantiated through experiments involving multiple saliency models across three classification datasets and various deep neural network (DNN) architectures.

**Strengths:**

(1)	The evaluation of interpretability methods at the pixel level provides a novel perspective, potentially offering more granular insights into model interpretation and improving the understanding of saliency interpretation methods.

(2)	The experimental validation is sound and complete, encompassing three widely-used datasets, commonly used DNN classifiers, and various advanced saliency detection techniques, thus ensuring a robust evaluation framework.

**Weaknesses:**

(1)	While the approach of assessing pixel-level saliency is innovative, the manuscript's experimental results do not seem to yield significant new insights or discoveries. This raises concerns about the overall contribution and significance of the proposed evaluation method.

(2)	The manuscript would benefit from additional visualizations which compare with the impacts of pixel-level perturbations across different interpretable methods. Such comparisons would more clearly illustrate the importance of pixel-level evaluation in interpreting significance.

**Questions:**

(1)	In Section 3.2, the manuscript discusses the use of Shapley values to estimate pixel importance. Would it possible to elaborate on how Shapley values effectively reflect the impact of some specific pixels on deep networks? Additionally, is it possible to compare with other methods in terms of the effectiveness on estimating pixel importance?

(2)	In Figure 2, the experiments add and remove important pixels to assess the accuracy of pixel estimation across different methods. It would be more insightful to understand the contributions of Shapley values versus other pixel importance estimation strategies in this context. A comparative analysis with the Shapley values curve would enhance the demonstration of the proposed method's effectiveness in pixel estimation.

(3)	The significant regions highlighted in Figure 3 look diverging from human perception, failing to adequately reflect areas of importance related to common "birds." It would be better providing additional labels for the samples and discussing potential reasons for the discrepancies in perception from human’s interpretations?

(4)	In Table 4, it is evident that gradient-based interpretability methods generally outperform perturbation-based methods, with notably less variances. Is it possible to provide an explanation on this phenomenon? This observation raises concerns about whether the proposed evaluation method effectively differentiates from the various interpretability approaches.

---

### Official Review · Reviewer_LAcm · 2024-11-04

**Soundness:** 4
**Presentation:** 3
**Contribution:** 2
**Rating:** 5
**Confidence:** 4

**Summary:**

The submission proposes a method to assess the correctness of XAI methods applied to DNN-based image classifiers. They propose to evaluate these methods at pixel level, by introducing a spatially localized shortcut to the image,  verifing the shortcut, and estimating the Shapley value for each pixel in the shortcut area in order to generate groundtruth explanation map. The proposed method is evaluated on 14 XAI methods on various dataset, with both ResNet-50 and ViT-B. The authors also compared the proposed framework to other XAI evaluation metrics to show its effectiveness.

**Strengths:**

1) more objective evaluation of XAI methods is a very important and unsolved problem.
2) using Shapley value to evaluate pixel importance is an interesting idea.
3) overall the proposed frame work sounds reasonable for evaluation of XAI model correctness

**Weaknesses:**

1) section 3, which is the most important section of the paper, is rather confusing. In 3.1 it is unclear how the "shortcut" was identified/introduced.
2) I am not sure equation (1) is correct or not. In order for the perturbed patch to be considered as "dominant area", shouldn't classifier f produce a lower probability on the perturbed image? Also shouldn't the classifier f predict the correct label on clean image and wrong label on the perturbed image?
3) the impact of using avg Monte Carlo sampling to substitute Shapley value should be evaluated in the context of this paper
4) evaluating the method on only ResNet and Vit-B doesn't seem convincing enough - especially since this is a method designed to be the "golden standard" for visual explanations.
5) (minor) the digits in the table seem very small while there are still space left at the end.

**Questions:**

1) line 200 on page 4, there is no x or x head in equation (1).
2) there are quite some design choices unclear to me - e.g. is there only one shortcut / dominant area?  if not how is this number determined? How are the k values determined in equation (3)?
3) how are the K and D values determined for different dataset?

---

### Note · Authors · 2024-11-14

I have read and agree with the venue's withdrawal policy on behalf of myself and my co-authors.